# CHOmics: A web-based tool for multi-omics data analysis and interactive visualization in CHO cell lines

Dongdong Lin[1], Hima B. Yalamanchili[1], Xinmin Zhang[2], Nathan E. Lewis[3,4], Christina S. Alves[1], Joost Groot[1], Johnny Arnsdorf[4], Sara P. Bjørn[4], Tune Wulff[4], Bjørn G. Voldborg[4], Yizhou Zhou[1]*, Baohong Zhang[1]*

1 Biogen Inc., Cambridge, Massachusetts, United States of America, 2 Bioinforx Inc., Madison, Wisconsin, United States of America, 3 Departments of Pediatrics and Bioengineering, University of California, San Diego, United States of America, 4 Novo Nordisk Foundation Center for Bio sustainability, Technical University of Denmark, Kgs. Lyngby, Denmark

☯ These authors contributed equally to this work.
* yizhou.zhou@biogen.com (YZ); baohong.zhang@biogen.com (BZ)

**Data Availability Statement:** All RNA-Seq data files are available in SRA and has the following BioProject ID: PRJNA613438" AVAILABLE at

## Abstract

Chinese hamster ovary (CHO) cell lines are widely used in industry for biological drug production. During cell culture development, considerable effort is invested to understand the factors that greatly impact cell growth, specific productivity and product qualities of the biotherapeutics. While high-throughput omics approaches have been increasingly utilized to reveal cellular mechanisms associated with cell line phenotypes and guide process optimization, comprehensive omics data analysis and management have been a challenge. Here we developed CHOmics, a web-based tool for integrative analysis of CHO cell line omics data that provides an interactive visualization of omics analysis outputs and efficient data management. CHOmics has a built-in comprehensive pipeline for RNA sequencing data processing and multi-layer statistical modules to explore relevant genes or pathways. Moreover, advanced functionalities were provided to enable users to customize their analysis and visualize the output systematically and interactively. The tool was also designed with the flexibility to accommodate other types of omics data and thereby enabling multi-omics comparison and visualization at both gene and pathway levels. Collectively, CHOmics is an integrative platform for data analysis, visualization and management with expectations to promote the broader use of omics in CHO cell research.

## Author summary

Recombinant proteins have dominated recent blockbuster therapeutic drugs, accounting for 11 of the top 15 drugs by sales. Chinese hamster ovary (CHO) cells are the most widely used expression system for biomanufacturing of many of these biotherapies. Thus, there is increasing interest in leveraging omics technologies for CHO cell line development, bioprocess optimization, and biotherapeutic product quality assessment. However, CHO cells have been largely ignored in the development of publicly available tools to facilitate

https://www.ncbi.nlm.nih.gov/bioproject/PRJNA613438.

**Funding:** The data used in this study was generated with support from a grant provided to the Technical University of Denmark by the Novo Nordisk Foundation (NNF10CC1016517) to N.E.L. The funders had no role in study design, data collection and analysis, decision to publish, or preparation of the manuscript.

**Competing interests:** The authors have declared that no competing interests exist.

comprehensive omics data analysis and management, despite being a ubiquitous research tool and biotherapeutic production host. To address the gap, we have recently developed a web-based tool, named "CHOmics", for the integrative and interactive data analysis and visualization specifically designed for CHO. This novel tool provides all-in-one solutions from raw data processing to pathway and gene analysis and offers considerable flexibility to customize analysis and visualization. It further allows for other omics data inputs and thereby enables multi-omics comparison. The open-source tool is freely available at http://www.chomics.org.

This is a *PLOS Computational Biology* Software paper.

## Introduction

With the increased usage of CHO cells in the large-scale production of pharmaceutical proteins, knowledge about the process optimization and biotherapeutic product quality becomes essential. Conventionally, cell line and cell culture process development are mostly based on empirical knowledge and statistical designs, and investigation of product quality deviation to identify the root cause often requires tremendous resources and time. More recently, omics and systems biology approaches have shown the potential to facilitate identification of predictive markers and the molecular mechanisms associated with various bioprocess phenotypes [1–3]. There are different omics technologies, each focused on a different biological question. While individual omics technologies have great utility for improving bioproduction in CHO, they are closely interconnected, and each can influence data interpretation from others. Therefore, analyzing data derived from multi omics technologies together will enable scientists to accurately predict and optimize cell culture aspects and further genetically modify cell lines.

Over the last decade, numerous studies have adopted high throughput omics-based approaches to elucidate CHO cell characteristics and the underlying cellular machineries. For example, several transcriptomic and proteomic studies have explored the relationship between gene expression and high production yield under varying culture conditions [4,5]. Despite this progress and relevant investigation, surprisingly few tools are available for data analysis and visualization of omics data in CHO cells. Although one recently developed open-source tool, PaintOmics [6], provides the ability to load transcriptomics and metabolomics measurements and visualize them over pathway maps, it requires input data to be pre-processed and normalized. There're also a few commercial packages available, however, they are typically costly, less flexible to customization, and requires proprietary databases. Moreover, many of the tools heavily rely on murine and human models, which makes it difficult to use them for CHO omics analysis. Because of these challenges, omics data processing and analysis often requires dedicated talent with tremendous time input.

With the improved Chinese hamster genome as reference (NCBI Refseq Annotation Release 103) [7], we established an integrated CHO-specific multi-omics platform, "CHOmics", that serves as a one stop-shop for omics data analysis from raw data to comparative pathway analysis across multiple omics data sets. As shown in Fig 1, the tool mainly consists of three modules including data input, analysis (preprocessing pipeline and statistical analysis)

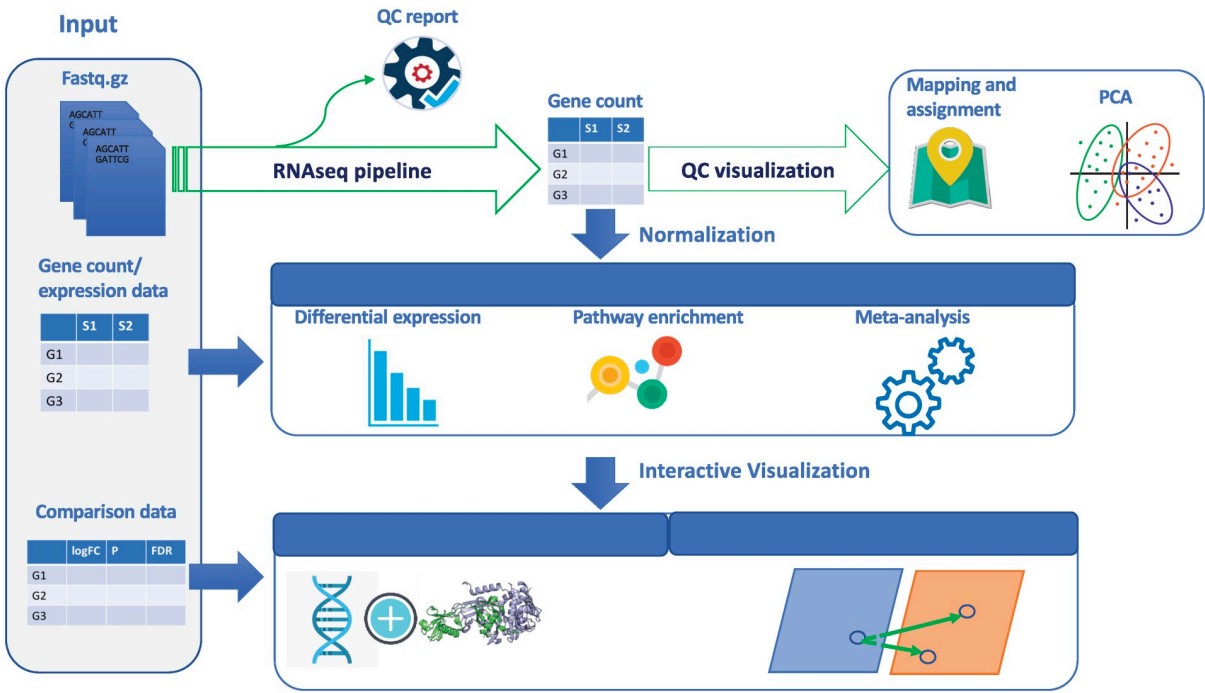

**Fig 1. The schematic view of CHOmics platform.** Different modules in the platform are shown encapsulating different functionalities like data input, data analysis using RNA-Seq pipeline, statistical analysis, and visualization.

and visualization. It is an open-source, user-friendly integrative analytical platform designed for biologists to analyze complex omics data with the capability of visualizing the analysis outputs interactively.

## Materials and methods

### Data input

CHOmics provides a flexible approach to allow multiple types of inputs including RNA sequencing (RNA-Seq) data and metadata from URLs, local folders, or remote servers. The data is organized in top-down structure with four levels including project, experiment, comparison, and sample.

**Transcriptomics data.** CHOmics has built in a comprehensive pipeline for RNA sequencing. Raw sequencing data (e.g., fastq or fastq.gz files) can be uploaded along with sample annotation as an experiment to be preprocessed by the pipeline. The analysis output can be imported to specific project for visualization and comparison.

**Gene-level data.** Gene level expression data (e.g., a count table or normalized expression data) preprocessed by external pipelines is accepted and subjected to further analysis in CHOmics. Various types of omics data can be presented at the gene level, such as transcriptomics from sequencing or microarray, proteomics, Ribo-Seq [8] or any other data type wherein a measurement that has a gene-level identifier can be mapped to a gene name. CHOmics accepts Entrez Gene IDs as gene identifiers which are further used to match gene ID from KEGG [9], Gene Ontology, Reactome [10] or WikiPathways [11] databases for pathway enrichment analysis.

**Comparison data.**   Comparison data are statistical outputs by comparing omics data between two conditions. It could be generated by internal pipeline or uploaded directly from external analysis. A statistical output table can include logFC, p-value, adjusted p-values, and other additional measures. By specifying an annotation file, users can easily link the summarized statistical outputs to the annotated samples, experiment, and project.

**Meta data.**   Besides the data imported for analysis, several meta data files describing the nature of an experiment (e.g., project name, platform, and disease, etc.) are necessary for sample annotation and management.

## Data analysis

CHOmics provides four analysis modules including: a built-in RNA-Seq data processing pipeline, differential expression (DE) analysis, functional pathway enrichment analysis, and meta-analysis, as shown in Fig 1. In each module, interactive plots are provided to enable comprehensive visualization of data and analysis results.

**RNA-Seq pipeline.**   Once raw RNA-Seq fastq files are uploaded, a preprocessing pipeline can be launched with the following steps: quality control, alignment and gene count generation.

*Quality control.* Fastq files are first evaluated for read quality by fastqc [12]. A summary table of fastqc output is generated for users to quickly check multiple properties of reads in each sample including per base sequence quality, content, per sequence quality scores, sequence length distribution, and overrepresented sequences.

*Alignment.* Reads after quality control are aligned to specified reference genome (e.g., Chinese hamster PICR genome, GCA_003668045.1 with NCBI Refseq Annotation Release 103: https://www.ncbi.nlm.nih.gov/genome/annotation_euk/Cricetulus_griseus/103/) by using the subread alignment tool [13]. Phred offset score and other mapping parameters (e.g., min votes, allowed mismatches, and max indels) are set for alignment. Junctions are also estimated during the alignment and summarized in the table along with read mapping metrics (e.g., mapping ratio and the number of detected gene, etc).

*Gene Count and normalization.* By comparing the aligned Bam files against the gene annotation file, CHOmics generates a gene count table by applying the 'featureCount' function in subread with specified strandedness. In addition, Trimmed Mean of M-values (TMM) normalization is applied to the raw counts to remove differences in the composition of the RNA population between samples. The normalized gene counts are then transformed to log2 scale using the voom method from the limma package for analysis and visualization [14].

As shown in Fig 2A, multiple plots are generated in the process for QC purpose. For example, the summary plot for mapping and gene assignment quality can help to identify samples with quality issues such as low total number of reads or genes, or low genome mapping rate. In addition, CHOmics enables the visualization of sample global expression profiling by using multidimensional hierarchical clustering plots and heatmap (Fig 2B), giving clear indication of sample similarity based on gene expression. Additionally, principle component analysis (PCA) empowers users to explore expression similarity among samples based on top variable genes or candidate gene set and provides a guidance for detecting potential outliers (Fig 2C). Users can interactively select Principal Components (PCs) to visualize the samples at different coordinates, and label them by different color, shape and size according to sample attributes.

**Differential expression analysis.**   The platform enables a statistical analysis of differential expression (DE) between conditions using gene count tables generated by aforementioned processing pipeline. A filtering step is allowed for removing low expressed genes by setting the cut-off for the count per million (CPM) and thereby reduces the burden of multiple hypothesis

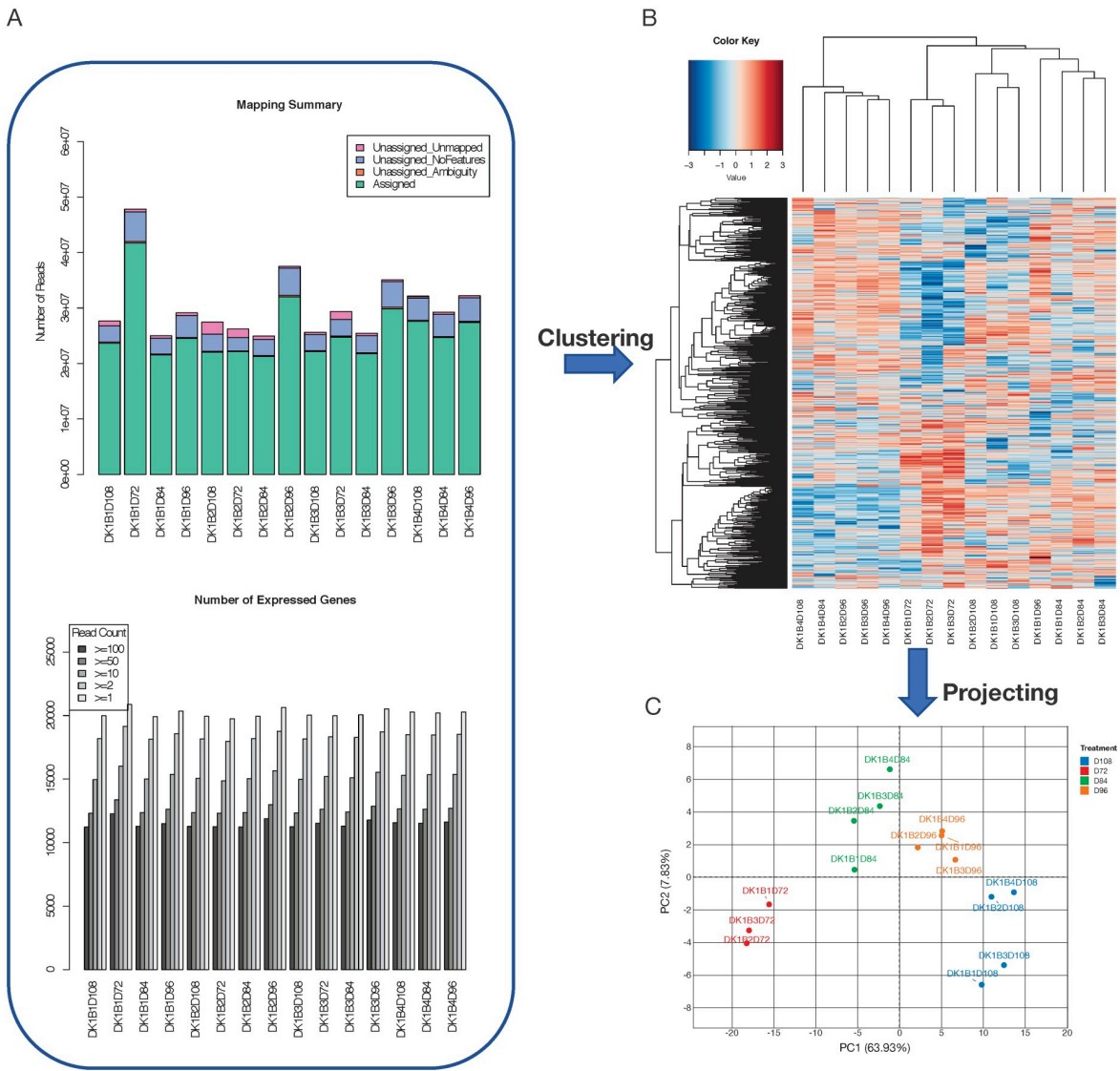

**Fig 2. Visualization of raw data processing output.** Gene mapping and expression distribution plots are shown (A) to check the sequencing reads processing quality and distribution. The samples can be (B) clustered based on their expression profiling or (C) subjected to principle component analysis to visualize expressional similarity among samples.

testing. The retained genes are normalized and log2-transformed followed by application of the linear model to the comparison between conditions using limma/voom package.

For each comparison, the statistics are reported including log fold change (logFC), p-value, and false discovery rate (FDR) corrected for multiple hypothesis testing with the Benjamini-Hochberg procedure. To highlight the differentially expressed genes (DEGs), CHOmics enables filtering of genes by FC and FDR values. In addition, CHOmics can either select those DEGs from a single comparison or select the common or pooled DEGs from multiple comparisons. This flexibility in gene selection enables users to focus on the characterization of candidate gene list across comparisons or projects. Based on the selected DEG list, the users can explore the heatmap of sample-gene expression and the volcano plot with both up- and down-regulated DEGs labelled, as shown in Fig 3A.

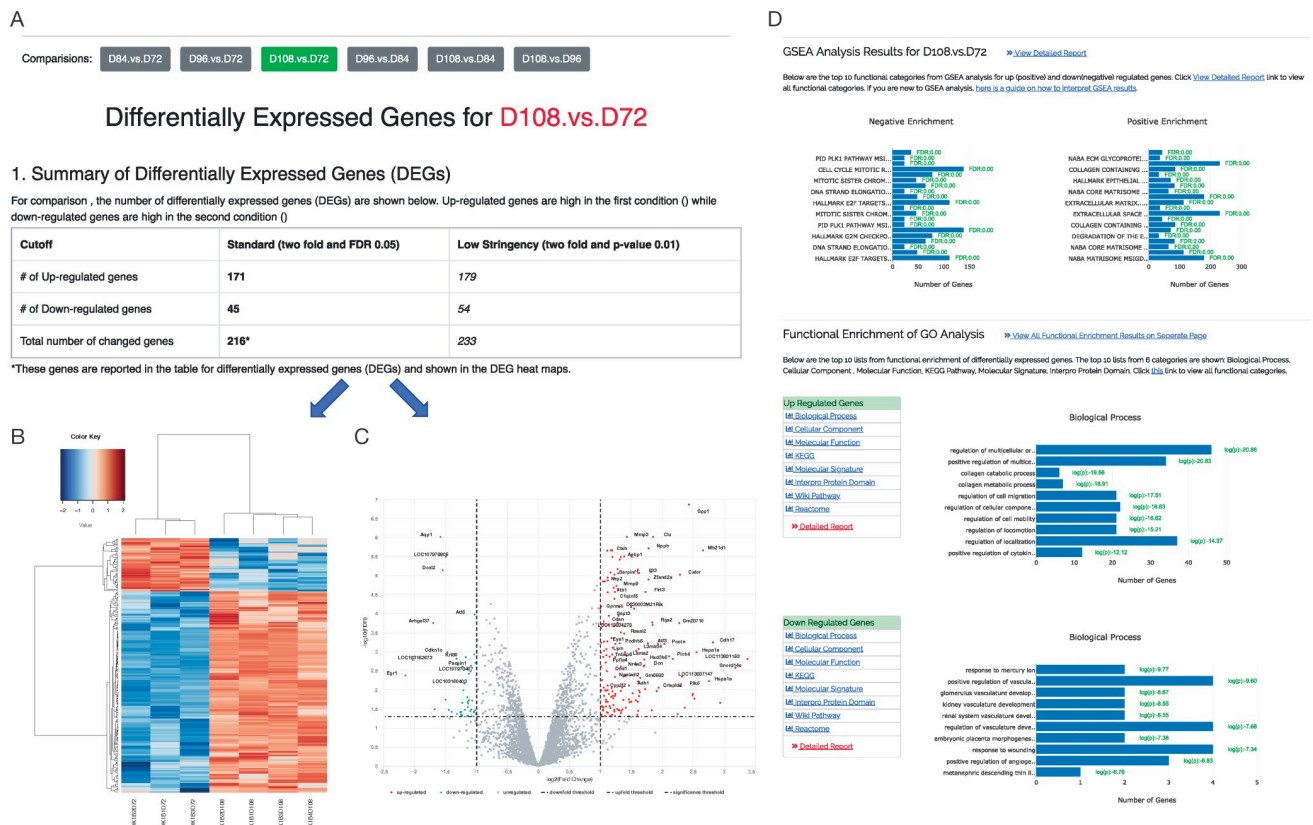

**Fig 3. The outputs from differential gene expression and pathway enrichment analysis.** (A) Differential expression analysis identified DEGs from the comparison between group D108 and D72 and are plotted in (B) heatmap and (C) volcano chart. (D) Pathway enrichment analysis showed the top 10 significant pathways from multiple databases enriched by DEGs.

**Pathway enrichment analysis.** Functional pathway analysis can be performed by both Gene set enrichment analysis (GSEA) and gene ontology (GO) enrichment methods in CHOmics. GSEA analysis tends to identify functional categories from CHO pathway database which are significantly overrepresented at the top or bottom of a ranked list of genes. The GO enrichment method uses an accumulative hypergeometric distribution model to test the over-representation of DEGs on pathways against all genes. The GO enrichment method is built on the Homer program [15] and multiple pathway databases such as Gene Ontology, KEGG Pathway, Molecular Signature, Interpro Protein Domain, WikiPathways and Reactome. Significantly enriched pathways are tested for the up- or down- regulated genes separately in each comparison as shown in Fig 3B. Bar-plots are also provided to show most significant pathways as well as the number of genes and the enrichment test p-values.

**Meta-analysis.** To increase the power of identifying DEGs across datasets, CHOmics provides a module to perform meta-analysis as illustrated in section 3.3.3 of supplementary tutorial by using diverse methods including Rank Product (RP), p-values combined by Fisher method, and p-values combined by maxP. The RP method is a non-parametric statistical test to detect genes that are consistently upregulated (or downregulated) among the projects. The p-value combining methods derive the combined p-value by using Fisher's combination or selecting the maximum p-value. CHOmics provides a summary plot of the significance of genes across projects by bubble plot to show the trends of gene expression changes across projects.

## Multi-omics and multi-layer visualization

One of the core modules in CHOmics is the interactive visualization tool that enables users to compare features across projects and omics at different levels (e.g., gene and pathway). The features to be viewed could be either a single gene or a list of genes (e.g., DEGs) and the samples to be compared could come from one project or across different projects.

**Multi-omics visualization.**   For a specified gene, CHOmics can plot the expression level of this gene across different omics data and under different conditions (e.g., time points) as shown in Fig 4A. Users can interactively evaluate the features by grouping and coloring the samples from different conditions. A set of genes can also be compared by employing hierarchical biclustering to explore intricated gene-sample relationship across omics (Fig 4B). In addition, to summarize the extent of gene expression changes, CHOmics can provide an overview of the fold change and significance of features (e.g., DEGs) derived from the statistical analysis across comparisons and omics as shown in Fig 4C.

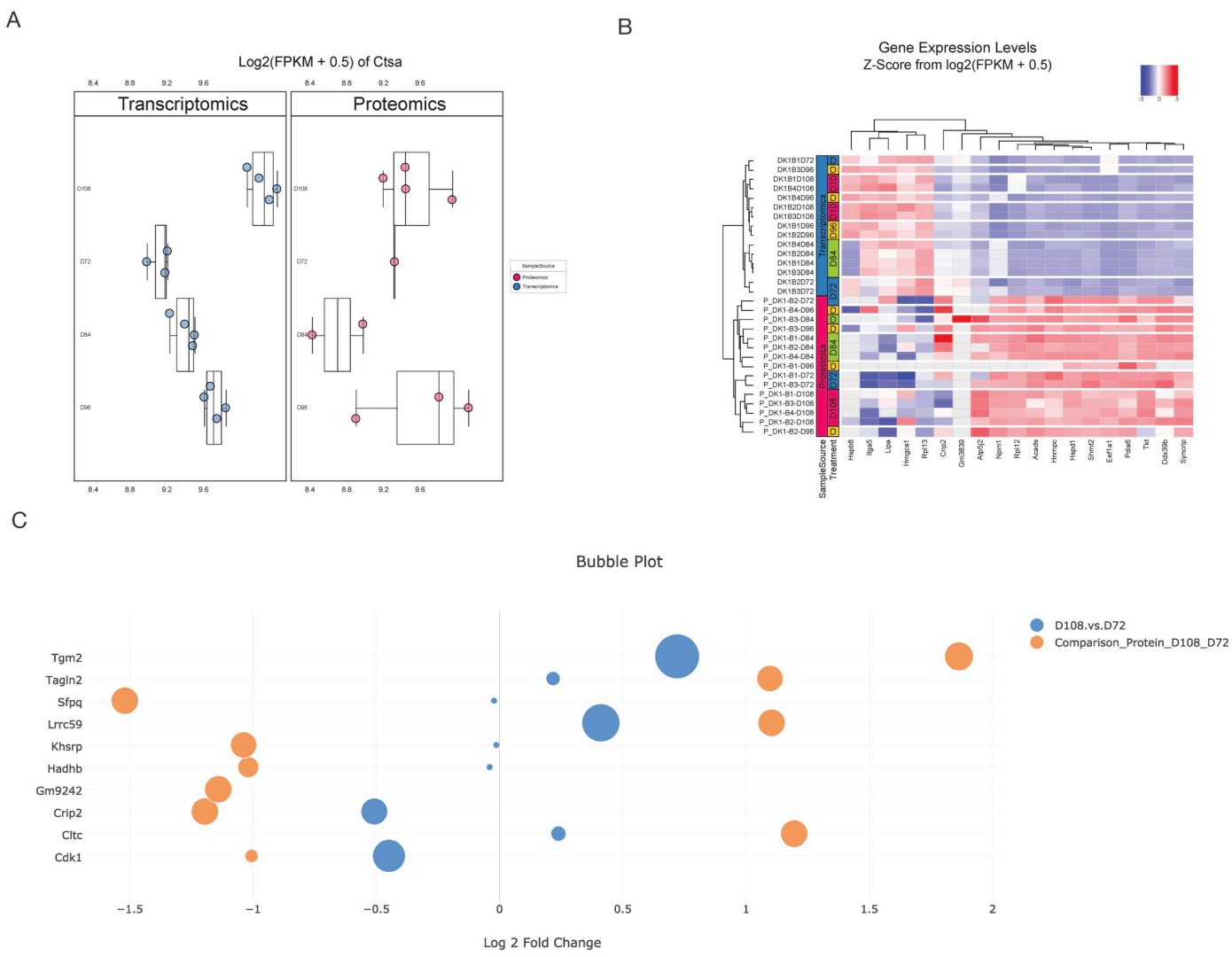

**Fig 4. The visualization of gene expression.** (A) Box plot of a gene or (B) Heatmap of a list of genes from different conditions and omics. (C) DEGs of interest can be visualized across comparisons and omics in a bubble plot.

**Multi-layer visualization.**   Besides multi-omics visualization of DEGs, CHOmics allows users to characterize the comparisons on pathways from multiple databases. Given the comparison data inputs selected from projects, CHOmics can generate a heatmap for top enriched pathways across comparisons. Users can check the heatmap intensity which indicates the enrichment significance, and other enrichment information (e.g., number of enriched genes), and then identify a specific pathway of interest for another layer of exploration (i.e., comparing gene level changes in the context of a pathway). Pathway diagrams show the pathway structure overlaid with the gene-level statistical results from different comparisons, demonstrating gene expression patterns among comparisons as well as their relationship to the other genes in the pathway.

## Results

### Use case demonstration

**Case1: Multi-omics analysis on profiling CHO-S cell growth.**   Here we demonstrate how to use CHOmics for analyzing multi-omics data (primarily transcriptomics and proteomics) from CHO cell lines. A Chinese Hamster Ovary-Suspension (CHO-S) clone was expanded and cultured. Starting at 72 hr into culture and every 12 hr thereafter to 108 hr, cells were harvested for transcriptomic analysis via RNA-Seq (pair-end 2x50bp) and proteomics analysis was conducted via mass spectrometry to identify genes differentially expressed from exponential growth to stationary phase (see [16] for details on omics data collection and preprocessing).

We first uploaded the RNA-Seq fastq files and initiated the built-in RNA-Seq pipeline. QC metrics reports are generated as shown in Fig 2A. Summary plots show that all the samples have moderate sequencing depth with at least 20 million reads, a high read mapping rate, and similar distribution in gene read counts. After read mapping, samples can be clustered based on gene expression profiles and variation can be further analyzed by PCA analysis. The PCA plot (Fig 2C) suggests that the samples are mainly clustered based on collection time points.

After completion of the pipeline, a gene count table was generated and normalized for differential expression analysis between the time points. Fig 3A lists DEG results from the comparison between 72 hr and 108 hr. 171 DEGs were significantly up-regulated at 108 hr, while 45 DEGs were down-regulated (FDR<0.05). The top DEGs with large effect size (absolute value of logFC > 1) are labeled in the Volcano plot (Fig 3C). For instance, high upregulation of the genes *CTSA* and *CTSB* at 108 hr indicates over-expression of these lysosome related genes at longer culture time [17]. Down-regulation of the gene early growth response protein 1 (EGR1) suggests reduction of this transcription factor which functions in cell growth and development [18]. In addition, identified DEGs can be further interpreted by pathway analysis as shown in Fig 3D. The analysis indicates that up-regulated DEGs are significantly enriched in some KEGG pathways related to cell development and cell death such as the lysosome, focal adhesion and apoptosis pathways.

Similarly, in proteomics analysis, after mapping protein ID to gene ID, we uploaded the protein measurement table and differential analysis results. PCA analysis on protein measures show that samples are clustered according to the time points (S1 Fig; one sample at 96 hr was excluded), which is in line with RNA-Seq results. The Volcano plot highlights multiple differential expressed proteins between time 108 hr and 72 hr in S2 Fig, including the up-regulation TGM2, which is implicated in the regulation of cell growth, differentiation, and apoptosis, and the down-regulation of SFPQ, which was reported to be critical for cell survival [19]. By overlapping DEGs from both omics analyses, we identified multiple genes with consistent changes across omics, including genes *TGM2*, *CRIP* and *CLTC*. Pathway analysis was also performed

and cross-checked with the results from transcriptomics analysis, showing some pathways consistently enriched by upregulated genes including those involved in the HIF-1 signaling pathway and down-regulated genes associated with the ribosome, glycolysis and gluconeogenesis (Fig 5A).

**Case2: Multi-omics profiling of three CHO parental host cell lines.** We used CHOmics to re-analyze transcriptomics and proteomics data from a study of profiling three commonly used parental cell lines (CHO-K1, CHO-DXB11, and CHO-DG44) in suspension cultures [20]. The transcriptomics data (RMA normalized log2 intensities) and proteomics data (normalized and scaled protein levels) were obtained from the paper. Differential analyses between the three CHO host cell lines were performed at both gene and protein levels using the R limma package. The expression matrices and comparison results were uploaded to the CHOmics. Ensembl ID for transcriptome and CHO gene symbols for proteome reported in the paper were recognized automatically by the CHOmics for gene mapping.

We demonstrated the reproducibility of CHOmics by performing PCA, differential analysis, and GO enrichment analysis across omics data sets. The results (S3–S5 Figs) are in good agreement with reported in the paper. Furthermore, CHOmics offers extra analysis and visualization functionalities, such as PCA analysis on the subset of genes (e.g., genes from specific pathway) as shown in S6 Fig, investigating the effect size of selected genes across multiple comparisons as shown in the bubble plot (S7 Fig), and enrichment analysis on multiple pathway databases (e.g., KEGG, WikiPathways) allowing users to map the differential analysis results from multi-omics data sets on any specific pathway (e.g., Glutathione metabolism) as shown in S8 Fig.

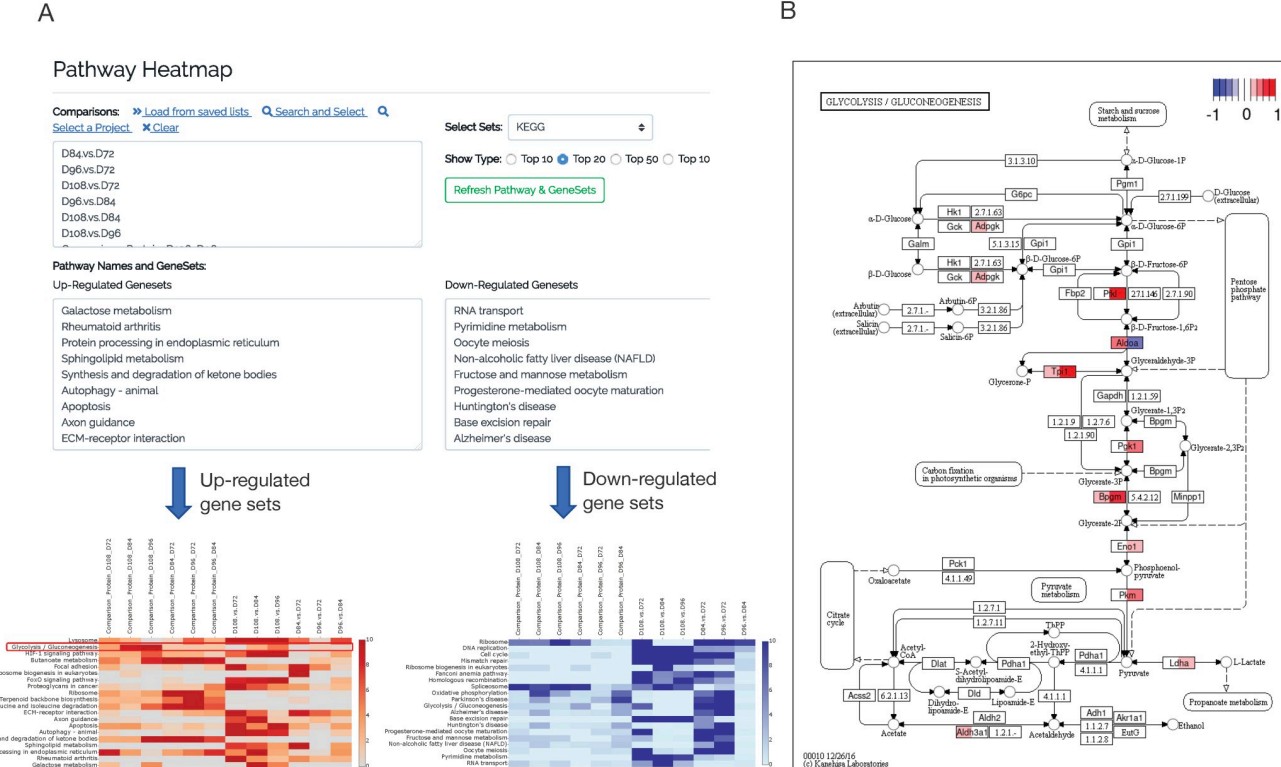

**Fig 5. Visualization of pathway enrichment across omics and comparisons.** By selecting comparisons, CHOmics can plot (A) heatmap for pathway enrichment across comparisons and (B) pathway diagram to show gene expression pattern from a specific pathway when involving multiple comparisons where the two colors in each node represent RNA and protein changes, respectively.

## Discussion

Here, we presented CHOmics platform for the integrative and interactive exploration of omics data from CHO cell lines. CHOmics is a web-based tool designed with considerable flexibility in analysis, visualization, and management of CHO omics data. Users can perform omics data analyses in a variety of ways through either launching the internal RNA-Seq pipeline to analyze raw data or uploading intermediate results from external pipelines. Versatile functionalities such as PCA and hierarchical clustering are provided to help users overview the data quality and distribution, and statistical analyses (e.g, DE analysis, pathway enrichment) to further explore the biological signals and interpretation. Moreover, CHOmics can summarize the analysis results across omics, comparisons and projects by meta-analysis to increase the feature detection power.

Another advantage of CHOmics is its ability to enable users to visualize data metrics and analysis results in an integrative and interactive way. Users can visualize the expression profiles of a gene or gene set across conditions or omics data sets, thus facilitating deeper understanding and interpretation of biological findings. Given the integrative capability, users can visualize the dynamics of omics data in response to conditions through time course analyses. Beyond gene level, CHOmics also provides a bird's-eye view of the functional pathways enriched by differentially expressed genes between biological conditions. Furthermore, CHOmics can map gene-level expression changes to pathway diagrams. Thus, this multi-layer visualization enables users to gain additional insights from colocalization of gene expression changes of multiple experiments on the same pathway.

Finally, CHOmics offers an effective way of managing projects from different sources such as internal or external data and/or analysis results. Along with flexibility in data input, CHOmics organizes data by hierarchical categories such as project, comparison, and samples. This centralized design makes comparison across projects at multiple levels (e.g., gene, sample and comparison) possible.

## Availability and future directions

CHOmics is free to use and is distributed under GPL license. The demo of client-side is available at http://chomics.org and has been extensively tested with Chrome and Firefox browser. Detailed tutorial can be accessed as supporting information and also available at https://bit.ly/2PyUxk5 in high resolution format. The source code written by multiple programming languages PHP, R and JavaScript, is available at https://github.com/baohongz/CHOmics. Installation procedure is provided at the link http://chomics.org/chomics/install.php. The demo site is installed on a dedicated server from source code mainly for visualization of results. If the users want to run the data preprocessing pipeline on a large-scale raw data, which usually requires significant computational resources, it is recommended to install the platform on a local server or create a Google cloud instance from publicly available Google Cloud machine Image "chomics-org20200806". CHOmics server-side application has been tested on Ubuntu and CentOS powered servers. Support for installing the system locally or in the cloud can be obtained by contacting info@bioinforx.com. Although the current version of CHOmics only contains a data processing pipeline for RNA sequencing, this is a continuous effort and more pipelines for other omics data will be incorporated in the future. In addition, the open-source platform can be extended to other species with minor configuration.

## Supporting information

**S1 Fig. Principle component analysis (PCA) on proteomics data.** (A) PCA analysis on proteomics data shows that one sample at 96 hr is outlier. (B) The samples are clustered mainly by

treatment (i.e., time points) after filtering out the outlier.
(TIFF)

**S2 Fig. Volcano plot on proteomics data.** The top differentially expressed proteins between 108 hr and 72 hr.
(TIFF)

**S3 Fig. Principle component (PC) analysis plots of transcriptomics and proteomics data from a study of profiling three commonly used parental cell in suspension cultures [20].** (A) Nine samples from three groups (CHO_DG44, CHO_Dukxb11, and CHO_K1) were clustered based on the first and second PCs of transcriptomics data. (B) The samples were clustered based on the first and second PCs of proteomics data.
(TIFF)

**S4 Fig. Venn diagram plots to show overlap of differentially expressed genes between CHOmics and reported from the paper [20] in both (A) transcriptomics and (B) proteomics data.**
(TIFF)

**S5 Fig. Gene Ontology (GO) enrichment analysis of differentially expressed genes from both transcriptomics and proteomics data.** The enrichment analysis on (A) biological processes, (B) molecular functions, and (C) cellular components of GO.
(TIFF)

**S6 Fig. PCA plots of transcriptomics data on subset of genes.** (A) The genes were selected from (A) N-glycan biosynthesis pathway, and (B) oxidative phosphorylation pathway of KEGG.
(TIFF)

**S7 Fig. Bubble plot of selected genes across comparisons and omics.** Common differentially expressed genes from comparisons of both transcriptomics and proteomics data analysis are shown. The bubble sizes are proportional to significance levels (-logFDR) of differentially expressed genes in various comparisons that are color-coded.
(TIFF)

**S8 Fig. Plots of KEGG pathway enrichment analysis.** (A) Top 20 pathways enriched by up-regulated differentially expressed genes from both transcriptomics and proteomics data. (B) Pathways enriched by down-regulated differentially expressed genes. (C) Differential analysis statistics from multi-omics data were aggregated into the pathway diagram of Glutathione metabolism from KEGG database. Each box is divided into equal stripes to show color-coded log2 fold changes capped at 1 where each stripe corresponds to one comparison.
(TIFF)

**S1 Text. CHOmics tutorial.**
(PDF)

## Author Contributions

**Conceptualization:** Dongdong Lin, Hima B. Yalamanchili, Xinmin Zhang, Yizhou Zhou, Baohong Zhang.

**Data curation:** Xinmin Zhang, Nathan E. Lewis, Johnny Arnsdorf, Sara P. Bjørn, Tune Wulff, Bjørn G. Voldborg.

**Formal analysis:** Dongdong Lin, Hima B. Yalamanchili, Xinmin Zhang.

**Funding acquisition:** Yizhou Zhou, Baohong Zhang.

**Investigation:** Dongdong Lin, Hima B. Yalamanchili, Xinmin Zhang, Nathan E. Lewis, Yizhou Zhou, Baohong Zhang.

**Methodology:** Dongdong Lin, Hima B. Yalamanchili, Xinmin Zhang.

**Project administration:** Yizhou Zhou, Baohong Zhang.

**Resources:** Xinmin Zhang, Nathan E. Lewis, Baohong Zhang.

**Supervision:** Yizhou Zhou, Baohong Zhang.

**Validation:** Dongdong Lin, Hima B. Yalamanchili, Nathan E. Lewis.

**Visualization:** Dongdong Lin, Hima B. Yalamanchili, Xinmin Zhang, Baohong Zhang.

**Writing – original draft:** Dongdong Lin, Hima B. Yalamanchili, Xinmin Zhang, Nathan E. Lewis, Christina S. Alves, Joost Groot, Yizhou Zhou, Baohong Zhang.

**Writing – review & editing:** Dongdong Lin, Hima B. Yalamanchili, Xinmin Zhang, Nathan E. Lewis, Christina S. Alves, Joost Groot, Johnny Arnsdorf, Sara P. Bjørn, Tune Wulff, Bjørn G. Voldborg, Yizhou Zhou, Baohong Zhang.

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
