## [Decision Letter · Decision Letter 0]

27 Jul 2020

Dear Dr. Zhang,

Thank you very much for submitting your manuscript "CHOmics: a web-based tool for multi-omics data analysis and interactive visualization in CHO cell lines" for consideration at PLOS Computational Biology.

As with all papers reviewed by the journal, your manuscript was reviewed by members of the editorial board and by several independent reviewers. In light of the reviews (below this email), we would like to invite the resubmission of a significantly-revised version that takes into account the reviewers' comments.

Please directly address the innovation concerns in your revision.

We cannot make any decision about publication until we have seen the revised manuscript and your response to the reviewers' comments. Your revised manuscript is also likely to be sent to reviewers for further evaluation.

Sincerely,

Jason A. Papin

Editor-in-Chief

PLOS Computational Biology

Reviewer's Responses to Questions

**Comments to the Authors:**

Reviewer #1: This is an exciting project. I spent my formative years in CHO informatics learning how to do all of the steps of this pipeline individually. This tool will streamline the CHO data analysis process and represents a huge time and monetary saving in terms of a researcher having the lack or expensive hardware to run these processes and not needing to devote countless hours of learning how to do this manually or hiring someone who can. The website is intuitive and walks the user through how every important process works and why it's used. Just this aspect alone is a great learning tool. I was impressed by the ease of use and the actual usefulness of the website.

I was able to reproduce the figures and data sets generated in this manuscript by downloading the sample fasta files and manually processing them. This manuscript successfully invents a way to lower the barrier of entry into CHO data analytics and is to be commend.

I would have liked to have seen some modularity in the kinds of tools available for use. For instance, some researchers would prefer using subread aligner while others might prefer STAR. But that is a minor quibble. The manuscript does a great job of laying the case for why this tool is important and delivers on the execution of the tool.

Reviewer #2: The authors designed a comprehensive web-based tool for analysis and visualization of CHO omics data. Especially, the tool could be useful in the visualization of different metabolic states of CHO cells, and thus practically relevant for users who are working with various omics data from CHO cell cultures. However, I can’t find any significance or novelty in terms of method development. I think the current work can be submitted to more relevant journals such as Bioinformatics or BMC Bioinformatics after front-end web application setup in the dedicated server so that users can directly do the analysis rather than installing the program. Open source should be available and accessible without signing.

I have some minor comments and questions for further improvement:

1. There are some abbreviations that are not stated clearly: CHO-S, FDR, TMM. The full names should be provided.

2. More details on requirements for installing CHOmics should be provided. Which web browser is most suitable or which programming language is needed to implement CHOmics source code?

**Have all data underlying the figures and results presented in the manuscript been provided?**

Reviewer #1: Yes

Reviewer #2: Yes

PLOS authors have the option to publish the peer review history of their article (what does this mean?). If published, this will include your full peer review and any attached files.

Reviewer #1: No

Reviewer #2: No
---

## [Decision Letter · Decision Letter 1]

9 Sep 2020

Dear Dr. Zhang,

Thank you very much for submitting your manuscript "CHOmics: a web-based tool for multi-omics data analysis and interactive visualization in CHO cell lines" for consideration at PLOS Computational Biology. As with all papers reviewed by the journal, your manuscript was reviewed by members of the editorial board and by several independent reviewers. The reviewers appreciated the attention to an important topic. Based on the reviews, we are likely to accept this manuscript for publication, providing that you modify the manuscript according to the review recommendations.

Please make note of the comments from Reviewer #2 on the need to demonstrate value of the software tool with relevant data which can be readily accessed through several public resources.

Sincerely,

Jason A. Papin

Editor-in-Chief

PLOS Computational Biology

[LINK]

Reviewer's Responses to Questions

**Comments to the Authors:**

Reviewer #1: Thank you for addressing my concerns.

Reviewer #2: Well done and explaination is now clear to me. But, the current results showing the omics data processing and visualization features based on available datasets is not sufficient enough to demonstrate its applicability and usefulness unless the authors like to publish this tool development efforts in Bioinformatics journals. At least one case study should be provided. The authors can use available datasets from any CHO multi-omics publications and show the consistent results and even additional analysis which can be done by CHOmics.

**Have all data underlying the figures and results presented in the manuscript been provided?**

Reviewer #1: None

Reviewer #2: Yes

PLOS authors have the option to publish the peer review history of their article (what does this mean?). If published, this will include your full peer review and any attached files.

Reviewer #1: No

Reviewer #2: No
---

## [Editor Report · Decision Letter 2]

6 Nov 2020

Dear Dr. Zhang,

We are pleased to inform you that your manuscript 'CHOmics: a web-based tool for multi-omics data analysis and interactive visualization in CHO cell lines' has been provisionally accepted for publication in PLOS Computational Biology.

Best regards,

Jason A. Papin

Editor-in-Chief

PLOS Computational Biology

Jason Papin

Editor-in-Chief

PLOS Computational Biology

---

## [Editor Report · Acceptance letter]

9 Dec 2020

PCOMPBIOL-D-20-00665R2 

CHOmics: a web-based tool for multi-omics data analysis and interactive visualization in CHO cell lines

Dear Dr Zhang,

I am pleased to inform you that your manuscript has been formally accepted for publication in PLOS Computational Biology. Your manuscript is now with our production department and you will be notified of the publication date in due course.

With kind regards,

Nicola Davies
